# Vitamin A Ameliorated Irinotecan-Induced Diarrhea in a Piglet Model Involving Enteric Glia Modulation and Immune Cells Infiltration

**DOI:** 10.3390/nu14235120

**Published:** 2022-12-02

**Authors:** Meng Li, Yonggang Huang, Huimin Jin, Daixiu Yuan, Ke Huang, Jing Wang, Bie Tan, Yulong Yin

**Affiliations:** 1College of Animal Science and Technology, Hunan Agricultural University, Changsha 410128, China; 2Key Laboratory of Agro-Ecological Processes in Subtropical Region, Institute of Subtropical Agriculture, Chinese Academy of Sciences, Changsha 410125, China; 3Hainan Institute, Zhejiang University, Sanya 572000, China; 4Department of Medicine, Jishou University, Jishou 416000, China

**Keywords:** vitamin A, retinoic acid signaling, diarrhea, enteric glia, macrophage

## Abstract

Vitamin A (VA) and its metabolite, retinoic acid (RA), play important roles in modulating intestinal mucosal immunity, yet little is known about their regulatory effects on enteric nervous system function. The study aims to explore the protective effects of dietary VA on diarrhea in a piglet model involving enteric glia and immune cell modulation. Twenty-eight weaned piglets were fed either the basal or VA (basal diet supplemented with 18,000 IU/kg VA) diet and with or without irinotecan (CPT-11) injection. CPT-11 induced increased diarrhea incidence, immune infiltration, and reactive enteric gliosis. A diet supplemented with 18,000 IU/kg VA ameliorated the adverse effects of CPT-11 on the gut barrier. VA reduced diarrhea incidence and attenuated enteric glial gliosis, immune cell infiltrations, and inflammatory responses of CPT-induced piglets. An in vitro experiment with 1 nmol/L RA showed direct protective effects on monocultures of enteric glial cells (EGCs) or macrophages in LPS-simulated inflammatory conditions. Furthermore, 1 ng/mL glial-derived neurotropic factors (GDNF) could inhibit M1-macrophage polarization and pro-inflammatory cytokines production. In summary, VA exerted protective effects on the intestinal barrier by modulating enteric glia and immune cells, perhaps enhancing epithelial recovery under CPT-11 challenge. Our study demonstrated that RA signaling might promote the roles of enteric glia in intestinal immunity and tissue repair, which provided a reference for the VA supplementation of patient diets.

## 1. Introduction

Delayed-onset diarrhea represents the main dose-limiting toxicity of irinotecan (CPT-11), a clinical drug for advanced colorectal cancer [1]. CPT-11-induced diarrhea has a reported overall incidence of 50~80%. Even low-grade diarrhea remarkably interfered with anticancer treatment [2]. Severe diarrhea caused significant dehydration, electrolyte imbalance, and nutritional deficiencies, which are associated with early death in cancer patients [3,4]. Intestinal tissue maintenance and repair rely on the integrated activity of diverse cell types, such as epithelial [5], immune [6], and stromal cells [7]. In recent years, increasing research has identified the important role of enteric neuroglia networks in gut integrity [8]. Enteric glial cells (EGCs), the major constituent of the enteric nervous system, have been found to be closely associated with the maintenance of intestinal barrier function [9,10]. Pigs and humans have many similarities in intestinal functionality and composition, and the minimum nutritional allowance for pigs and humans is also comparable, making the pig an ideal animal model [11].

Vitamin A (VA) is an essential nutrient required for the maintenance of normal growth and development, visual function, immune function, and epithelial integrity [12,13]. Dietary VA is absorbed by intestinal epithelium cells and oxidized into retinoic acid (RA) in the tissues [14,15]. As an effective regulatory molecule, RA in target cells binds to RA receptors (RARs) and regulates Serum amyloid A protein (SAA) promoters to control gene transcription [16]. RA signaling acts on different cells of both innate and adaptive immune systems and maintains immune homeostasis [17]. VA deficiency leads to an impaired immune barrier and increased infection rates [18]. Studies show that dietary VA supplements could improve host defense and reduce mortality and morbidity in humans and animals [19,20]. Here, our research aimed to explore the roles of dietary VA on diarrhea incidence, intestinal immunity, and enteric glia function in a piglet model under irinotecan challenge.

## 2. Materials and Methods

### 2.1. Reagents and Chemicals

Irinotecan (CPT-11) was obtained from Hubei Chushuo Biotechnology Co., Ltd. (Jingmen, China). Pig IL-1β, IL-4, IL-10, IL-17, IL-22, TNF α, TGFβ, GDNF, SIgA, and lysozyme ELISA kits were obtained from Wuhan Huamei Biological Engineering Co., Ltd. (Wuhan, China). Mouse IL-1β, GDNF, and COX-2 ELISA kits were obtained from Wuhan Huamei Biological Engineering Co., Ltd. (Wuhan, China).

### 2.2. Animals and CPT-Induced Experimental Diarrhea of Piglets

The animal experiments were approved by the Institutional Animal Care and Use Committee of the Institute of Subtropical Agriculture, Chinese Academy of Sciences (2013020).

A total of 28 Yorkshire × Landrace × Duroc weaned piglets (21-d-old) with an average body weight (BW) of 7.29 ± 0.28 kg were randomly divided into four groups, including CON (basal diet), VA (basal diet +18,000 IU/kg VA), CPT-11 (basal diet), and VA + CPT-11 (basal diet +18,000 IU/kg VA) group with 7 piglets per group. The basal diet was designed to meet the nutrient requirements (National Research Council, 2012) for weaned piglets. To induce experimental diarrhea, on d 24, piglets of the CPT and VA + CPT groups received an intraperitoneal injection of CPT-11 (15 mg/kg BW) for 4 days. The doses of VA and CPT-11 were chosen according to the preliminary experiment on the basis of growth performance and diarrhea scores of the piglets, respectively. All piglets were individually housed in identical environments and allowed free access to food and water and were regularly observed for diarrhea, inappetence, and other abnormal behavior such as huddling and shivering (Figure 1). The whole experimental period lasted for 30 d. On d 30, all piglets were slaughtered after 12 h fasting. Colon samples were collected and immediately stored at −80 °C or fixed in a formaldehyde solution. The weight gain and feed intake of each group were calculated on the basis of the average value of each piglet.

### 2.3. Cell Culture and Treatment

Rat enteroglial cell line CRL-2690 (EGCs) and mouse macrophage cell line RAW 264.7 were purchased from American Type Culture Collection (ATCC, Manassas, VA, USA). The cells were cultured in Dulbecco’s Modified Eagle’s Medium (DMEM)–high glucose (containing 10% FBS, 100 U/mL penicillin, and 100 μg/mL streptomycin) and maintained at 37 °C in a humidified incubator of 5% CO_2_.

EGC were seeded in 6-well plates and cultured in the medium containing 0 or 1 nmol/L RA (Sigma-Aldrich, Burlington, MA, USA) and 0 or 1 μg/mL LPS (*E. coli* 055: B5, Sigma-Aldrich). After 24 h incubation, the supernatants were collected to quantify IL-1β production. The adherent EGCs were harvested for total RNA extraction and total protein extraction.

Rat RAW 264.7 cells were cultured in 6-well plates with medium containing 0 or 1 nmol/L RA (Sigma-Aldrich), 0 or 1 ng/mL GDNF (Sigma-Aldrich), and 0 or 1 μg/mL LPS (*E. coli* 055: B5, Sigma-Aldrich). After 16 h culture, the supernatants were collected to measure IL-1β and COX-2 production. The adherent macrophages were harvested for total RNA extraction.

### 2.4. Morphological Examination

After dehydration, colon samples were embedded in paraffin and cut into 5 um fractions using a microtome. The sections were further stained with hematoxylin and eosin (H&E) for morphological analysis. Images were acquired using Leica laser microdissection systems (DM6B, Heidelberg, Germany). Villous height and crypt depth were measured with computer-assisted microscopy (Micrometrics TM; Nikon ECLIPSE E200, Tokyo, Japan).

### 2.5. RNA Extraction and Quantitative Real-Time Polymerase Chain Reaction

Total RNA was extracted from colonic tissues, EGCs, or macrophage cells using TaKaRa Universal RNA Extraction Kit (Beijing, China), and then reverse transcribed by real-time qPCR was performed on a real-time PCR system (Applied Biosystems, Foster City, CA, USA). Primers (shown in Table 1) were designed with Primer 5.0 (PREMIER Bio soft International, Palo Alto, CA, USA) according to the gene sequence of the pig. β-actin and Gadph were used as two housekeeping genes to normalize target gene transcript levels.

### 2.6. Western Blot (WB) Analysis and Enzyme-Linked Immunosorbent Assay (ELISA)

Colon tissues or EGCs were dissolved in radioimmunoprecipitation assay (RIPA) buffer containing proteinase inhibitor and sonicated on ice, and then centrifuged at 14,000× *g* rpm for 20 min at 4 °C. The concentration of the total protein was determined by a BCA protein assay kit. For the WB assay, equal amounts of proteins (50 μg) were electrophoretically separated in a 10% sodium dodecyl sulfate-polyacrylamide gel (SDS-PAGE) and transferred to PVDF membranes. The membranes were subsequently incubated with blocking solution, followed by incubation with mouse anti-AMPK, anti-ERK, anti-p38, anti-Akt, and anti-β-actin primary antibodies overnight at 4 °C, and then incubated with relevant horseradish peroxidase-conjugated secondary antibodies at room temperature for 1 h. The proteins were finally visualized using Enhanced Chemiluminescence (CW0049A, CWBIO, China; RPN2235, GE Healthcare, Chicago, IL, USA) and X-ray Films (XBT-1, Kodak, Rochester, NY, USA). For the ELISA assay, cytokine concentrations in colon tissues or the supernatants of EGCs and macrophage cells were detected using porcine or rat IL-1β, IL-4, IL-10, IL-17, IL-22, TNFα, TGFβ, GDNF, SIgA, lysozyme ELISA kits according to the manufacturer’s instructions. The concentration of the total protein in the homogenates was measured by a BCA protein assay kit (Beyotime Institute of Biotechnology, Shanghai, China) to normalize the cytokine concentration.

### 2.7. Immunofluorescence

Paraffin-embedded colon tissues were deparaffinized in xylene and rehydrated through graded alcohol to water. After being permeabilized in blocking solution (10% fetal bovine serum and 1% Triton X-100, in PBS) for 2 h, whole microdissected tissues were incubated with anti-GFAP, anti-CD3, and anti-CD68 primary antibodies overnight at 4 °C, followed by FITC-conjugated secondary antibodies, and then counterstained with DAPI to stain the nuclei. Images were collected on a Leica TCS SPS microscope (Wetzlar, Germany).

### 2.8. Statistical Analysis

The data in this article were presented as means ± standard error of the means (SEM). All data were analyzed by GraphPad Prism 7.0 software (La Jolla, CA, USA) using an unpaired Student’s *t*-test or one-way ANOVA with Dunnet’s multiple comparisons test. *p*-values of less than 0.05 were taken to indicate statistical significance.

## 3. Results

### 3.1. Dietary VA Reduced Diarrhea Incidence and Attenuated Intestinal Damage Induced by CPT-11

Increased diarrhea incidence and damaged intestinal morphology were observed in piglets following CPT-11 injection. As shown in Figure 1A, CPT significantly decreased average daily weight gain (ADG) and increased diarrhea incidence. The H&E-stained images in Figure 1B,C exhibited the decreased villous height and villous height/crypt depth ratio in the jejunum and ileum of CPT-11-induced piglets. In the colon, CPT-11 caused crypt dilation and increased the expression of apoptosis protein AKT (Figure 1E). Dietary supplementation with VA increased weight gain (Figure 1A) and inhibited villous atrophy in the jejunum and ileum (Figure 1B,C) as well as crypt dilation and cell apoptosis in the colon (Figure 1D,E), indicating the protective effects of VA on growth and gut function following CPT-11 exposure. However, there was no significant difference in the growth and intestinal morphology of piglets between the CON group and the VA group. Dietary VA is absorbed by intestinal epithelium cells and oxidized into RA in the tissues [14,15]. We, therefore, detected the expression of RA signaling-related genes, Rara and Saa. As demonstrated in Figure 2A, CPT-11 significantly reduced Rara mRNA expression, while VA restored the Rara gene level in the CPT-11-challenged piglets. However, both VA and CPT-11 significantly increased the relative mRNA expression of SAA (Figure 2A). These results suggested dietary VA exerted protective effects on growth and the intestinal barrier in piglets.

### 3.2. VA Suppresses Immune Cell Infiltration and Inflammatory Responses

The intestinal immune system initiated gut defense and subsequently led to inflammatory responses and tissue repair upon CPT-11 exposure. As shown in Figure 3, the expression of CD3 and CD68 proteins were increased in CPT-11-induced piglets, which were obviously decreased while supplemented with dietary VA (Figure 3C,D), indicating that VA could suppress the CD3^+^ T cell and macrophage infiltration caused by CPT-11. Compared to the CON group, the levels of inflammatory genes *p38*, *p65*, *Il6*, *Il1b*, *Cox2*, *Il17*, *Cebpa*, and *Cebpb* were much higher in diarrhea piglets, which were decreased in the CPT + VA group. Furthermore, inflammatory cytokines production, including IL-1β and IL-17, was also decreased upon VA treatment. VA inhibited the expression of inflammatory cytokines but promoted the production of IL-22 (Figure 2A,B), which was involved in tissue repair. In summary, these results suggested that VA could suppress immune cell infiltration and inflammatory responses and promote tissue repair in CPT-induced piglets.

### 3.3. VA Inhibits Reactive Enteric Gliosis and Modulates Neuropeptides Production Following CPT-11 Challenge

To uncover the effects of VA on EGC function and neuropeptide production, immunofluorescence staining was performed. We first identified reactive enteric gliosis in CPT-11-induced piglets, referring to the reduced GFAP+ and S100β+ positive cells (Figure 3A) while VA treatment ameliorated reactive enteric gliosis. Furthermore, CPT-11 significantly increased GDNF production in CPT-11-treated piglets, which was decreased upon VA supplementation (Figure 3B). These results indicated that VA could inhibit reactive enteric gliosis and modulate neuropeptide production.

### 3.4. RA Regulates the Functions of Macrophages and EGCs In Vitro

A rat EGC cell line (CRL2690) and a rat macrophage cell line (RAW 264.7) were included in the present study to uncover whether RA could regulate the functions of macrophages and EGCs in vitro. As shown in Figure 4, neurogenic inflammation was observed in LPS-stimulated EGCs, referring to the significant increase in IL-1β (Figure 4C) and GDNF (Figure 4B) production and related gene (Il1b, Gdnf, and Bdnf) expression (Figure 4A). RA significantly inhibited the up-regulation of the Il1b and Gdnf genes as well as IL-1β and GDNF proteins in LPS-induced EGCs. As shown in Figure 5D,E, LPS-induced M1-macrophage polarization, evidenced by the up-regulated expression of the Cox2 and Il1b genes (Figure 4D) and the COX-2 and IL-1β cytokines (Figure 4E), and RA could suppress M1-macrophage polarization and related inflammatory responses.

### 3.5. GDNF Modulates Macrophage Functions In Vitro

To gain insights into the communications between EGCs and macrophages, GDNF was added to macrophage cultures. As shown in Figure 5, GDNF significantly suppressed the up-regulation of Il1b, il6, Inos, Tnfa, Cox2 genes, and IL-1β and COX-2 cytokines in LPS-stimulated macrophages, indicating that GDNF could suppress M1-macrophage polarization and inflammatory responses.

## 4. Discussion

CPT-11 is a clinical drug for advanced colorectal cancer [1,2,21] which induces bowel mucosal damage, significant dehydration, electrolyte imbalance, and nutritional deficiencies in cancer patients [1,22]. Previous studies have shown that CPT-11 induced intestinal pathological changes (such as villus atrophy, crypt expansion, and hypoplasia) and cell apoptosis in the rat [23]. In the present study, we use weaned piglets as an animal model because of their disturbed gut environment, which is similar to the gut of cancer patients. Our results show that the intraperitoneal injection of CPT-11 contributes to body weight loss and serious intestinal damage in piglets. These negative effects are mainly attributed to SN38, the active metabolite of CPT-11, which directly induced mucosa damage, mucosa hypersecretion, and the malabsorption of water and electrolytes [24]. Dietary VA is an important nutrient required for growth and intestinal homeostasis. A previous study reported that VA significantly increased the average daily gain (ADG) in weaned piglets [25]. In the mouse diarrhea model, retinol could also enhance intestinal adaptive immune responses against Clostridium difficile TxA challenge and alleviate intestinal damage [26]. In addition, VA enhanced the proapoptotic effects of irinotecan in cancer cells in vitro [27]. Our results demonstrated that VA decreased the intestinal apoptosis level and alleviated body weight loss and intestinal pathological changes in this research, and these protective effects are associated with RA metabolism.

VA is absorbed by intestinal epithelial cells and converted into RA, which interacts with RARs and binds directly to SAA promoters to regulate target gene transcription [13,26]. VA–RA signaling is essential for the maintenance of immune homeostasis during inflammation. Studies have shown that RA regulated gut-associated lymphoid tissue development and immune cell activation through RARα on the cellular membrane [28,29]. RARα was critical to the development of the intestinal immune system, and the specific deletion of RARα in intestinal epithelial cells resulted in an underdeveloped immune system and impaired epithelial barrier [29]. In this research, CPT reduced the relative expression of RARα, while VA significantly increased the RARα expression of the CPT-challenge piglets, suggesting RA signaling is impaired by CPT challenge and restored following VA supplementation. SAAs are immunoregulatory proteins that integrate dietary and microbiota signals by the intestinal epithelium. SAAs circulate retinol after bacterial exposure and promote Th17 response [30,31]. In the present study, both VA and CPT treatments upregulated *Saa* expression. Furthermore, the infiltration of CD3^+^ T cells and the expression of IL-17 were enhanced in the CPT-11 group but inhibited in the CPT-11 + VA group, indicating that VA might restore intestinal permeability and modulate immune responses via RA signaling.

Intestinal barrier dysfunctions, including increased epithelial permeability, defective immune responses, and increased inflammatory mediators are thought to contribute to the development of gut disease. In piglet diarrhea models, CPT-11 induced severe mucosa damage accompanied by immune cell infiltration and colonic inflammation [32]. In the present research, the increased expression of cytokine IL-1β and inflammatory genes p38, p65, Il6, Il1b, and Cox2 as well as the decreased expression of IL-22 appeared in the CPT-11-challenged piglets. Immunofluorescent staining of inflamed colon sections further revealed macrophages and T cell infiltration in the mucosal layers. These results were in accordance with previous findings that CPT-11 promoted macrophage infiltration into intestinal tissue through p65 NF-κB and p38 MAPK signaling pathways, resulting in a robust IL-1β response and colonic inflammation [33,34]. IL-22, an IL-10 family member, contributes to intestinal-epithelium resistance to injury and is involved in gut defense and tissue repair [35]. We found VA promoted IL-22 production and suppressed macrophage infiltration as well as related inflammatory gene and cytokine expression, indicating VA could inhibit colonic inflammation and promote tissue repair in CPT-11-challenged piglets. Correspondingly, RA could suppress the polarization of naïve macrophage to M1-macrophage and related inflammatory responses in vitro.

The enteric nervous system (ENS) is known as the “little brain of the gut” and has been implicated in the regulation of gut inflammatory conditions. EGCs, the major constituent of CNS identified by the expression of markers GFAP and S100β, play an important role in the maintenance of mucosal barrier function [36]. During intestinal inflammation and infection, EGCs become “reactive” (reactive gliosis) and undergo changes in morphology, GFAP expression, and cytokine release, which contribute to neuroinflammation and barrier dysfunction [37]. Cytokine GDNF, for instance, was upregulated in IBD patients and involved in the initial stage of colonic inflammation [35]. A recent report demonstrated that a reduction in reactive gliosis could advance the treatment of intestinal diseases with inflammation or neuroinflammation [37]. In CPT-11-induced piglet diarrhea, an obvious increase in GFAP and S100β as well as GDNF was found in the present research, and VA inhibited the reactive gliosis and neuroinflammation in the colon tissue. Moreover, our vitro experiment indicated that LPS initiated immune responses in EGCs and upregulated GDNF production to induce protective responses. Consistently, our in vitro experiment showed RA could inhibit the protective responses in LPS-stimulated EGCs. However, the mechanisms were unclear and would be further explored in the following research.

Neuropeptides, such as GDNF, BDNF, and substance P, are secreted by mucosal EGCs and have significant roles in modulating intestinal inflammation and gut homeostasis [38]. Several neuropeptides could directly function in immune responses and barrier integrity. Among these, GDNF emerges as a novel number in the family of protective mucosal factors in colonic inflammation. It has been suggested that EGCs sense microenvironment cues and control GDNF production to regulate type 3 innate lymphoid cells and gut defense in the colon [39]. Moreover, GDNF in DSS-induced mice was also found to prevent enteric neuron loss and reduce colonic inflammation [40]. Intestinal macrophages have crucial roles in inflammation development and tissue homeostasis [41]. The tissue microenvironment provides signals that can direct the polarization of monocytes towards a pro-inflammatory (M1) macrophage or towards a pro-resolving (M2) macrophage [41]. Bacterial LPS or cytokine IFN-γ could induce the polarization of a naïve macrophage into an M1 macrophage, which in turn produces high levels of proinflammatory cytokines such as IL-1β, TNF, IL-6, and COX-2 to facilitate gut immunity and defend foreign pathogens [42,43]. Our piglet experiment showed that EGCs and macrophages were involved in the regulatory effects of VA on gut homeostasis and host health. Though the dose of vitamin A in our study shows protective effects on the intestinal barrier, it may have negative health effects in a longer study. This is a possible shortcoming of our study. In the vitro experiment, we use LPS to build the M1 macrophage models. Notably, we first found GDNF could inhibit M1 macrophage polarization and related inflammatory responses in vitro, indicating that EGC–macrophage interactions might also be involved in the regulation of intestinal inflammation and gut homeostasis.

## 5. Conclusions

Our study demonstrated that VA supplementation exerted protective roles in CPT-11-induced diarrhea involving enteric glia modulation and immune cell infiltration. In vivo, VA alleviated diarrhea incidence and intestinal damage, suppressed immune cell infiltration, and inhibited inflammation responses. In vitro, RA directly modulated the functions of EGCs or macrophages in LPS-simulated inflammatory conditions. GDNF could inhibit M1 macrophage polarization and related inflammatory responses in vitro, highlighting the impact of neuro-immune response on intestinal mucosal damage and providing an encouraging mechanism of VA in modulating intestinal neurogenic inflammation.

## Data Availability

Not applicable.

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
