# Peer review of "Vitamin A Ameliorated Irinotecan-Induced Diarrhea in a Piglet Model Involving Enteric Glia Modulation and Immune Cells Infiltration"

_nutrients, 2022, doi:10.3390/nu14235120_

Round 1

Reviewer 1 Report

The work by Meng et al. entitled “Vitamin A ameliorated irinotecan-induced diarrhea in a piglet model involving enteric glia modulation and immune cells infiltration” has major limitations in terms of data presentation and interpretation, which must be addressed in order to be considered for publication.

Major concerns:

1.     Materials and Methods. RAW 264.7 is a mouse cell line, but it was cited as rat cell line. Please confirm the rat-specific qPCR primers also work with the mouse sequence. It is highly unlikely that all the sequences will match by chance. This is a major flaw that raises questions regarding data interpretation, and has to be convincingly addressed.

2.     Figure 1E:

i)                 Please mention (or include in the figure) the n numbers used for the graphical representation (relative intensity).

ii)               The graphical representation and the western data shown do not match. For example, p38 level does not seem to be induced (rather appears to be downregulated) by CPT.

iii)              AMPK appears to be progressively lower from left to right; CPT+VA looks considerably lower than CON band, whereas graphs show no difference between them.

iv)              Which are the right bands for Akt and ERK?

v)               Please include the molecular wt. markers in the western blot figures.

3.     Lines 159-160: “VA exerted protective effects on 159 intestinal damage through its active metabolite, RA.” – no direct evidence shown. Please modify the statement.

4.     “However, both VA and CPT-11 significantly increased the relative 163 mRNA expression of SAA (Fig. 2A)” – Please provide insight what this means when Rara expression is reduced by CPT and restored in CPT+VA.

5.     “As shown in Fig. 4, the expression 173 of CD3 and CD68 protein were increased in CPT-11-induced piglets, which were obviously decreased while supplementation with dietary VA (Fig. 4C and D),” : Please correct the figure number. Please include a quantitative representation for each set from at least 3 replicates showing statistical significance.

6.     Please include the rectangle legends for the experimental conditions for all figures.

7.     Please consistently mention n number for all the figures. 

Minor/editorial

1.     Abstract. Line 18: first two sentences do not introduce the work well.

2.     Abstract. Please include full forms of the following abbreviations: VA, EGC

3.     Introduction. Line 44: Please include the underlined word “found to be closely”

4.     Introduction. Line 45: Pigs and humans have many similarities in intestinal functionality and composition (Q: not sure what ‘composition’ is referring to. Please be more specific) and the minimum nutritional allowance ..

5.     Introduction. Line 52: “SA A promoter”. write full form

6.     Materials and Methods. Line 75:  was were received

7.     Materials and Methods. Line 79: access to feed (Please replace with ‘food’) and water

8.     Materials and Methods. Line 80: “other abnormal behavior”. Please mention a few examples

9.     Materials and Methods. Lines 86 and 95: RAW 264.7 is a mouse cell line

10.  Section 3.1.Lines 151-2: Please specifically mention the Figure no. e.g. Fig. 1B and C.

11.  Figure 1B: As shown in Fig 1C, please include the rectangle legends for the experimental conditions.

12.  Figure 1D: Please include the graphical representation of the colon data as well.

13.  There are multiples instances of erroneous figure citation. For example, line 156: Fig. 2A and B would be Fig 1A and B?

Author Response

Thanks for your comments. We have explained and corrected these comments (seen as the following text).

Major concerns:

Ponint 1: Materials and Methods. RAW 264.7 is a mouse cell line, but it was cited as rat cell line. Please confirm the rat-specific qPCR primers also work with the mouse sequence. It is highly unlikely that all the sequences will match by chance. This is a major flaw that raises questions regarding data interpretation, and has to be convincingly addressed.

Response 1: Thanks for your comments. In line 86, we have revised it to mouse macrophage cell line RAW 264.7. The rat-specific qPCR primers did work with the mouse suquence. We are sure because we have repeated this experiment for at least three times.

Point 2: Figure 1E:

  1. i) Please mention (or include in the figure) the n numbers used for the graphical representation (relative intensity).
  2. ii) The graphical representation and the western data shown do not match. For example, p38 level does not seem to be induced (rather appears to be downregulated) by CPT.

iii)  AMPK appears to be progressively lower from left to right; CPT+VA looks considerably lower than CON band, whereas graphs show no difference between them.

  1. iv) Which are the right bands for Akt and ERK?
  2. v) Please include the molecular wt. markers in the western blot figures

Response 2: Thank you. We have corrected.

Point 3: Lines 159-160: “VA exerted protective effects on 159 intestinal damage through its active metabolite, RA.” – no direct evidence shown. Please modify the statement.

Response 3: Thank you. We have revised it to “Dietary VA is absorbed by intestinal epithelium cells and oxidized into RA in the tissues” and cited related refference.

Point 4: “However, both VA and CPT-11 significantly increased the relative 163 mRNA expression of SAA (Fig. 2A)” – Please provide insight what this means when Rara expression is reduced by CPT and restored in CPT+VA

Response 4: Thank you. We have provided insight when Rara expression is reduced by CPT and restored in CPT+VA. Line 252-255: In this research, CPT reduced the relative expression of Rara, while VA significantly increased Rara expression of CPT-challenge piglets, suggesting RA signaling is impaired by CPT challenge and restored following VA supplementation.

Point 5: “As shown in Fig. 4, the expression 173 of CD3 and CD68 protein were increased in CPT-11-induced piglets, which were obviously decreased while supplementation with dietary VA (Fig. 4C and D),” : Please correct the figure number. Please include a quantitative representation for each set from at least 3 replicates showing statistical significance.

Response 5: Thank you. We have corrected the figure number.

Point 6: Please include the rectangle legends for the experimental conditions for all figures.

Response 6: Thank you. We have added the rectangle legends for all figures.

Point 7: Please consistently mention n number for all the figures.

Response 7: Thank you. We have added n number for all figures.

Minor/editorial

Point 1: Abstract. Line 18: first two sentences do not introduce the work well.

Response: Thank you. We have modified.

Point 2: Abstract. Please include full forms of the following abbreviations: VA, EGC

Response: Thank you. We have added full forms of VA and EGCs.

Point 3: Introduction. Line 44: Please include the underlined word “found to be closely”

Response: Thank you. We have corrected.

Point 4: Introduction. Line 45: Pigs and humans have many similarities in intestinal functionality and composition (Q: not sure what ‘composition’ is referring to. Please be more specific) and the minimum nutritional allowance ..

Point 5: Introduction. Line 52: “SAA promoter”. write full form

Response: Thank you. We have added full form.

Point 6: Materials and Methods. Line 75:  was were received

Response: Thank you. Corrected.

Point 7: Materials and Methods. Line 79: access to feed (Please replace with ‘food’) and water

Response: Thank you. We have corrected.

Point 8: Materials and Methods. Line 80: “other abnormal behavior”. Please mention a few examples

Response: Thank you. We have added a few examples of abnormal behavior.

Point 9: Materials and Methods. Lines 86 and 95: RAW 264.7 is a mouse cell line

Response: Thank you. We have corrected.

Point 10: Section 3.1.Lines 151-2: Please specifically mention the Figure no. e.g. Fig. 1B and C.

Response: Thank you. We have corrected.

Point 11: Figure 1B: As shown in Fig 1C, please include the rectangle legends for the experimental conditions.

Response: Thank you. We have corrected.

Point 12: Figure 1D: Please include the graphical representation of the colon data as well.

Response: Thank you. We have added the graphical representation of the colon data.

Point 13: There are multiples instances of erroneous figure citation. For example, line 156: Fig. 2A and B would be Fig 1A and B?

Response: Thank you. We have corrected.

Reviewer 2 Report

This is interesting, well executed experimental study on protective effect of pretreatment with Vitamin A against irinotecan (anticancer drug frequently used for chemotherapy of colon cancer) - induced diarrhea. Authors suggest that positive results of their study could be applicable in clinical settings to prevent life threatening side effects of anticancer treatment with irinotecan. I have the following suggestions and questions listed below:

1. Material an methods - please add (for clarity) the figure - flow chart of study design.

2. Discussion: due to the fact that results of this study have possible application in clinical practice authors should discuss the following questions;

a/ colon cancer typically affects older adults. Are weaned piglets suitable experimental model for these human epidemiological data ?

b/ can simultaneous administration of Vitamin A change (suppres) anticancer activity  of irinotecan ?

c/ please compare daily and cumulative doses of Vitamin A used in animal experiments with those tolerable by humans (possible overdosing)

d/ are concentrations of Vitamin A used in in vitro experiments comparable to  those that can occur in human plasma ?

Author Response

Thank you for your comments. We have corrected and explained these comments.

Comments and Suggestions for Authors

This is interesting, well executed experimental study on protective effect of pretreatment with Vitamin A against irinotecan (anticancer drug frequently used for chemotherapy of colon cancer) - induced diarrhea. Authors suggest that positive results of their study could be applicable in clinical settings to prevent life threatening side effects of anticancer treatment with irinotecan. I have the following suggestions and questions listed below:

  1. Material an methods - please add (for clarity) the figure - flow chart of study design.

Response 1: Thank you. We have added.

  1. Discussion: due to the fact that results of this study have possible application in clinical practice authors should discuss the following questions;

a/ colon cancer typically affects older adults. Are weaned piglets suitable experimental model for these human epidemiological data ?

b/ can simultaneous administration of Vitamin A change (suppres) anticancer activity  of irinotecan ?

c/ please compare daily and cumulative doses of Vitamin A used in animal experiments with those tolerable by humans (possible overdosing)

d/ are concentrations of Vitamin A used in in vitro experiments comparable to  those that can occur in human plasma ?

Response 2: Thank you! Your comments are very instructive for our article.

a/ It’s difficult to select old pigs for our experiment. So we use weaned piglets as an animal model because of their disturbed gut envioronment, which is similar to the gut of cancer patients. (Line 251-252)

b/ Vitamin A has been reported to enhance the proapoptotic effects of irinotecan in cancer cells in vitro. But we didn’t focus on this point in the current reaearch. (Line 261-262)

c/ Though this dose of vitamin A shows protective effects on intetinal barrier, but it may have negative health effects in a longer study. This is a possible shortcoming of our study. (Line 335-337)

d/ Vitamin concentration of human plasma is about 2.8 umol/L. In our study ,we use 1 nmol/L retinoic acid in in vitro experiment.

Round 2

Reviewer 1 Report

Thanks to Meng et al for revising the manuscript.  A few concerns still remain to be addressed. 

a. Major concern: Fig 1E: p38 western data is not convincing. From the blot presented it does not look like it is induced by CPT, and CPT+VA treatment brings back the expression to basal levels. The authors need to provide more convincing evidence in support of their claim.

b. Regarding the rat-specific primers for the mouse RAW 264.7 cells, please include a statement.  

c. Fig 1E: The right bands for Akt and ERK were not indicated. 

Author Response

A few concerns still remain to be addressed.

  1. Major concern: Fig 1E: p38 western data is not convincing. From the blot presented it does not look like it is induced by CPT, and CPT+VA treatment brings back the expression to basal levels. The authors need to provide more convincing evidence in support of their claim.

Response: Thank you for your concern of p38 western data. We checked the original data and find that p38 protein expression had no significant changes among the four groups in our study. We thank you again and corrected it in our manuscript. All original data are presented here:

                    C1     A1     CPT1  CPT+A1  C2     A2    CPT2   CPT+A2

                   C3      A3      CPT3    CPT+A3

  1. Regarding the rat-specific primers for the mouse RAW 264.7 cells, please include a statement.

Response: Thank you. We added primers for the mouse RAW 264.7 cells.

  1. Fig 1E: The right bands for Akt and ERK were not indicated.

Response: Thank you. We added arrow to indicate the KDa of the bands.

Reviewer 2 Report

I am satisfied with authors responses and manuscript revision. 

Author Response

Thank you for your kindly comments.